# Quantifying the risk of spillover reduction programs for human health

**Scott L. Nuismer** [1]*, **Andrew J. Basinski**[2], **Courtney L. Schreiner**[3], **Evan A. Eskew**[2], **Elisabeth Fichet-Calvet**[4], **Christopher H. Remien**[5]

**1** Department of Biological Sciences, University of Idaho, Moscow, Idaho, United States of America,
**2** Institute for Interdisciplinary Data Sciences, University of Idaho, Moscow, Idaho, United States of America,
**3** Department of Ecology and Evolutionary Biology, University of Tennessee, Knoxville, Tennessee, United States of America, **4** Bernhard Nocht Institute for Tropical Medicine, Hamburg, Germany, **5** Department of Mathematics and Statistical Science, University of Idaho, Moscow, Idaho, United States of America

* snuismer@uidaho.edu

**Data Availability Statement:** All code and data used in the manuscript is available at: https://github.com/snuismer/Unanticipated-Consequences-of-Spillover-Reduction.

## Abstract

Reducing spillover of zoonotic pathogens is an appealing approach to preventing human disease and minimizing the risk of future epidemics and pandemics. Although the immediate human health benefit of reducing spillover is clear, over time, spillover reduction could lead to counterintuitive negative consequences for human health. Here, we use mathematical models and computer simulations to explore the conditions under which unanticipated consequences of spillover reduction can occur in systems where the severity of disease increases with age at infection. Our results demonstrate that, because the average age at infection increases as spillover is reduced, programs that reduce spillover can actually increase population-level disease burden if the clinical severity of infection increases sufficiently rapidly with age. If, however, immunity wanes over time and reinfection is possible, our results reveal that negative health impacts of spillover reduction become substantially less likely. When our model is parameterized using published data on Lassa virus in West Africa, it predicts that negative health outcomes are possible, but likely to be restricted to a small subset of populations where spillover is unusually intense. Together, our results suggest that adverse consequences of spillover reduction programs are unlikely but that the public health gains observed immediately after spillover reduction may fade over time as the age structure of immunity gradually re-equilibrates to a reduced force of infection.

## Author summary

Many pathogens, such as rabies, coronaviruses, and hantaviruses primarily circulate within wild animals but can infect humans when the opportunity arises. This pervasive challenge to public health has motivated the development of new methods designed to reduce the frequency of these spillover events. Although reducing spillover infection of humans appears to be an obvious win for public health, it is conceivable that altering historical patterns of spillover could change the age structure of human immunity in a way that undermines human health. Using mathematical and computational models we

**Funding:** Funding for this work was provided by National Institutes of Health grant 2R01GM122079 to SLN and CHR and by National Science Foundation grant DEB 2314616 to SLN. The funders had no role in study design, data collection and analysis, decision to publish, or preparation of the manuscript.

**Competing interests:** I have read the journal's policy and the authors of this manuscript have the following competing interests: S.L. Nuismer is listed as an inventor on a pending patent for a Lassa virus vaccine.

evaluate the conditions required for these counterintuitive impacts to occur. Our analyses demonstrate that reducing spillover from wild animals will generally improve public health and that negative outcomes can occur in only rare and unusual circumstances. Although negative impacts of spillover reduction are likely to be rare, our results show that the public health benefits of spillover reduction may fade over time unless a near total elimination of spillover can be achieved.

## Introduction

Zoonotic diseases, those that transmit from animals to people, are a pervasive threat to public health. Nearly two-thirds of emerging infectious diseases have a zoonotic origin [1], and chronic spillover of zoonotic pathogens sickens hundreds of thousands of people each year. Just within the past decade, spillovers of Ebola virus [2], Lassa virus [3], Nipah virus [4], and, most notably, SARS-CoV-2 [5–7] have collectively infected and killed millions and driven massive societal disruption [8, 9]. Clearly, mitigating the risk posed by spillover of zoonotic diseases is a pressing public health challenge.

Historically, our response to zoonotic disease has primarily focused on identifying and containing human infections and outbreaks before they spiral out of control [10, 11]. The mixed track record of this approach has led to increasing interest in preemptive strategies that seek to minimize the risk of spillover itself. For example, human behavior-change initiatives [12, 13] and ecological or technological countermeasures to disease [14–17] aim to reduce the force of spillover into the human population. As long as reducing the force of spillover reliably reduces the burden of zoonotic disease, these efforts represent a clear win for public health.

A critical assumption that underpins these preemptive approaches, however, is that the force of spillover and disease severity are independent. One scenario in which spillover pressure and disease severity become intertwined arises when the severity of pathogen infection increases with age [18]. In such cases, once established as endemic, the cumulative health burden of zoonotic disease may be lower in areas where spillover pressure is high. This counterintuitive result arises if high spillover pressure causes most humans to be infected at a young age when the health impacts of infection are less severe. If these individuals are exposed again later in life, they are then protected by immunity acquired in childhood. In contrast, in areas with low spillover pressure, infection may not occur until later in life when its consequences are more severe. The relationship between age of first infection and disease severity forms the basis for existing theory linking changes in vaccination rates or disease transmission to shifts in the public health burden of established human diseases [19–23]. These ideas have not yet, however, been formally integrated into our theoretical understanding of the link between spillover pressure of zoonotic pathogens and human disease. Thus, we lack a quantitative framework for predicting how changes to reservoir populations that influence spillover pressure (e.g., habitat destruction, habitat reconstruction, displacement, culling, vaccination) will influence the total public health burden of zoonotic disease.

Here, we seek to clarify if spillover reduction, whether intentional or unintentional, could increase the overall public health burden of zoonotic disease. To this end, we develop mathematical models of age-structured populations subject to a constant force of spillover. Our models apply to those zoonotic diseases caused primarily by repeated, direct spillover from an animal reservoir rather than an initial spillover followed by sustained human-to-human transmission. Using mathematical analyses and computer simulations, we study how the public health burden of disease will change in response to shifts in the force of spillover. These

analyses identify the conditions where counterintuitive impacts of spillover reduction arise and quantify their magnitude. Parameterizing our models using empirical data on Lassa virus allows us to evaluate the scope for spillover reduction to cause unanticipated and undesirable consequences in a system with significant impacts on human health.

## Methods

### Continuous-time model

We study an age-structured human population that experiences a constant force of infection, $\lambda$, caused by spillover from an animal reservoir. We assume human infection results directly from spillover and that human-to-human transmission following spillover is sufficiently rare to be ignored. This assumption is reasonable for zoonotic pathogens such as Lassa virus, *Brucella*, and *Borellia* where most human infection is the result of direct spillover from an animal reservoir [3, 14]. Humans are born into the susceptible class, $S$, at rate $b$ and die at a per capita rate $\delta$. Individuals who become infected through spillover move into an infected class, $I$, from which they may recover at rate $\gamma$ or advance to a state of clinical disease, $M$, at rate $\mu$. Individuals experiencing clinical disease, $M$, may recover at rate $\gamma$ or succumb to the disease at a per capita rate $v$. Individuals that recover from infection move to the $R$ class which is assumed to be immune to further infection. However, immune individuals in the $R$ class may lose immunity over time and return to the susceptible $S$ class at rate $\omega$. Assuming the human population is sufficiently large for stochastic effects to be ignored leads to the following system of partial differential equations describing the distribution of humans of age $a$ in each class as a function of time, $t$:

$$\frac{\partial S}{\partial t} = -\frac{\partial S}{\partial a} - \lambda S + \omega R - \delta S \tag{1a}$$

$$\frac{\partial I}{\partial t} = -\frac{\partial I}{\partial a} + \lambda S - \gamma I - \delta I - \mu I \tag{1b}$$

$$\frac{\partial M}{\partial t} = -\frac{\partial M}{\partial a} + \mu I - \gamma M - \delta M - v M \tag{1c}$$

$$\frac{\partial R}{\partial t} = -\frac{\partial R}{\partial a} + \gamma(I + M) - \omega R - \delta R \tag{1d}$$

with boundary conditions:

$$S(0, t) = b \tag{2a}$$

$$I(0, t) = 0 \tag{2b}$$

$$M(0, t) = 0 \tag{2c}$$

$$R(0, t) = 0 \tag{2d}$$

In the equations above, the functional dependence of the variables on age, $a$, and time, $t$, has been suppressed for clarity.

Because our focus is on understanding how changes to the force of spillover influence human health at the population level, we focus on quantifying the "burden" of zoonotic disease which we define as the number of new clinical cases that occur each year, per capita.

Mathematically, the clinical burden of zoonotic disease at time, $t$, is then given by:

$$\mathcal{B} = \frac{1}{N} \int_0^\infty \mu(a) I \, da \qquad (3)$$

For generality, we also quantify the public health impacts of spillover reduction by studying changes in average human lifespan.

## Stochastic simulations

Because the continuous-time model developed in the previous section is largely intractable for many biologically interesting cases, we complemented it with a simulation-based approach. In addition to expanding the scope of scenarios we can investigate, these simulations allow us to probe the temporal dynamics of the system during transitions from a state of high spillover pressure to one where spillover pressure has been reduced. Specifically, we use the Gillespie algorithm to simulate stochastic analogues of equations (1–2) with discrete age classes. We use this stochastic approach to better quantify uncertainty in outcomes, particularly changes in the burden of clinical disease which emerge from spillover—a potentially rare event associated with significant randomness. To increase computational efficiency, we implement a version of the $\tau$-leaping algorithm [24] with a time-step equal to 1 day. Transitions between age classes are discrete and deterministic and occur once each year when all individuals are advanced into the next age class. To speed computation and enhance biological realism, we impose a maximum lifespan of 150 years on these stochastic simulations such that individuals older than 150 years inevitably die and are removed from the system. Stochastic simulations were implemented in C++.

## Application: Lassa virus in West Africa

To better understand the relevance of our results for a system of significant importance to human health, we parameterized our model for Lassa virus within West Africa. Lassa virus (*Lassa mammarenavirus*; family Arenaviridae) is a single-stranded bisegmented RNA virus that is endemic throughout much of West Africa where it imposes a substantial burden on human health due to regular spillover from its primary rodent reservoir, *Mastomys natalensis* [3, 25]. Spillover occurs primarily through human contact with urine or feces of infected rodents although isolated human-to-human transmission is also possible, primarily in nosocomial settings [25]. For this reason, spillover reduction has been explored as a means to reduce the health burden of this widespread disease [26, 27]. Accumulating evidence also suggests that an invasion by the black rat, *Rattus rattus*, may be displacing *Mastomys natalensis*—and reducing the force of spillover—as it moves inland from the coast [28–31]. Thus, understanding the consequences of spillover reduction in this system is relevant to predicting the consequences of intentional human intervention as well as incidental anthropogenic impacts.

Estimates for most model parameters were gleaned directly from the literature, with values and sources described in the S1 Text. In brief, disease-independent mortality, $\delta$, was estimated as the reciprocal of the average lifespan of humans from Ghana, Guinea, Sierra Leone, Liberia, and Mali. The rate of recovery from infection, $\gamma$, was based on an average duration of infection equal to 30 days which we selected based on the frequently reported 7–21 day pre-symptomatic period of infection [32] and 18 day average symptomatic period of infection [33]. Our estimate for the rate of progression to clinical disease, $\mu$, was derived from values in the literature reporting that 20% of infections result in development of symptoms [32]. Similarly, our estimate for the rate of progression to death once symptoms develop, $\nu$, was derived from values in the literature reporting that 30% of symptomatic cases admitted to the hospital result in

mortality [32]. We set the rate at which immunity wanes, $\omega$, to the rate of seroreversion reported by McCormick et al. 1987 [34]. Because considerable uncertainty and debate surrounds the duration of immunity, however, we also consider an alternative scenario where immunity is permanent. As we will see, negative impacts of spillover reduction are more likely with permanent immunity, making this a useful (conservative) benchmark for evaluating potential risks of planned interventions.

To understand the relationship between age and disease severity within the Lassa virus system, we analyzed survival data for individuals in different age classes admitted to the hospital with clinically-verified Lassa virus infections [35]. Specifically, we calculated the proportion, $P_i$, of individuals within each age class, $i$, that succumbed to Lassa virus infection. These proportions were than transformed to age-class specific rates of advancing to severe disease, $\mu_i$, by recognizing that:

$$\mu_i = \frac{P_i(\gamma + \delta)}{1 - P_i} \tag{4}$$

Least squares regression was then used to estimate a linear function predicting the rate of advancing to clinical disease as a continuous function of age, $\mu(a)$.

Finally, we estimated the force of spillover into the human population, $\lambda$, for sites within West Africa where the seroprevalence of Lassa virus antibodies has been systematically estimated within the human population. Assuming the population is at a steady state, the force of spillover can be estimated using the following equation:

$$\hat{\lambda} = -\frac{\mathcal{R}(\delta + \omega)(\gamma + \delta + \mu)(\gamma + \delta + \nu)}{(\gamma + \delta + \mu + \nu)(\gamma(\mathcal{R} - 1) + \mathcal{R}(\delta + \omega))} \tag{5}$$

where $\mathcal{R}$ is the estimated seroprevalence of Lassa virus antibodies (S1 Text).

## Results

### Decreasing the force of spillover increases the average age of infection

Decreasing the force of spillover reduces the number of people being infected per unit time and should thus lead to an intuitive decrease in disease within the human population. In addition to decreasing incidence, however, reducing the force of spillover increases the average age at which individuals become infected. Although this relationship is well-understood within classical epidemiology [36], its consequences for spillover reduction programs have not been fully explored. Thus, to set the stage for results we will develop in the next section, we calculate the average age at infection for our model and describe how it changes with the force of infection and duration of immunity.

Because we assume the force of spillover is independent of age, the average age at infection is equal to the average age of individuals in the susceptible class, $S$:

$$\bar{\mathcal{A}} = \int_0^\infty a \frac{S(a)}{\hat{S}} \, da \tag{6}$$

where $S(a)$ is the number of susceptible individuals of age $a$ at steady state, and $\hat{S}$ is the equilibrium abundance of susceptible individuals (S1 Text). Assuming disease-dependent mortality, $\nu$, is negligible, Eq (6) can be solved to yield the following expression for the average age of infection:

$$\bar{\mathcal{A}} = \frac{\delta^2(\gamma + \delta)^2 + \omega^2((\gamma + \delta)^2 + \gamma\lambda) + \omega(\gamma\lambda(\gamma + 2\delta) + 2\delta(\gamma + \delta)^2)}{\delta(\gamma + \delta)(\delta + \omega)(\gamma(\delta + \lambda + \omega) + (\delta + \lambda)(\delta + \omega))} \tag{7}$$

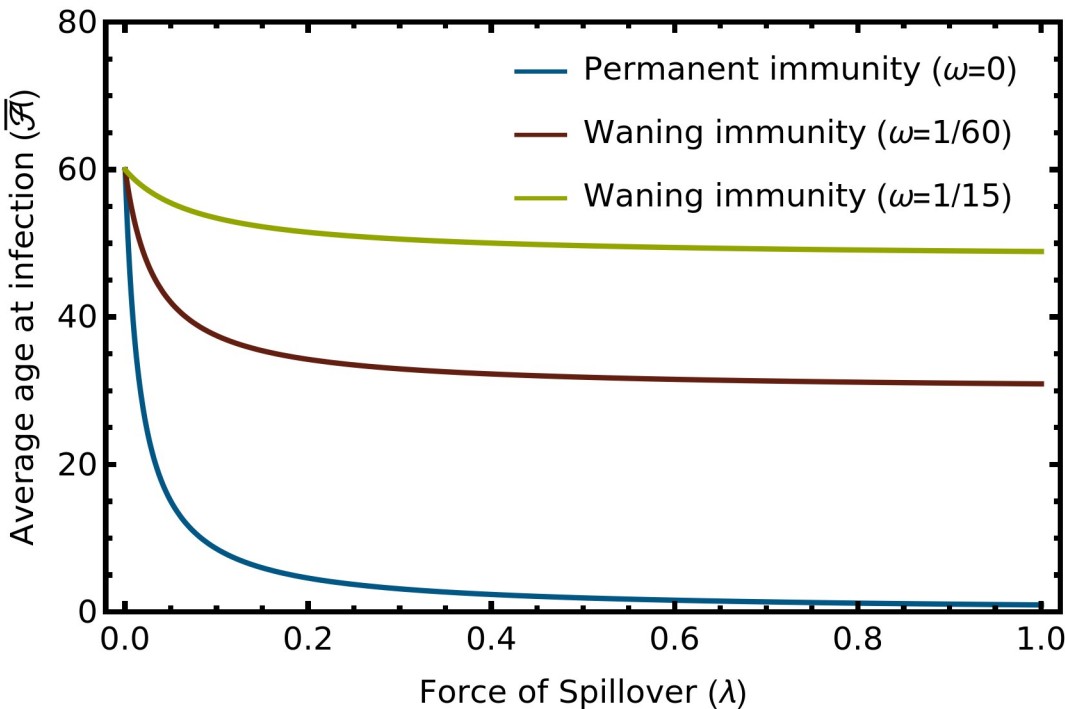

**Fig 1. The relationship between the force of spillover (λ) and the average age at infection $\bar{\mathcal{A}}$.** The blue line shows the case of permanent immunity, the read line immunity that lasts, on average, for the expected lifespan of the human population, and the green line immunity that lasts for, on average, only 1/4 of the expected human lifespan. The remaining parameters were: $\mu$ = 2.89, $\delta$ = 1/60, $\nu$ = 0, $\gamma$ = 365/14.

Numerical exploration of expression (7) reveals two important results (Fig 1). First, as the force of spillover falls, the average age at infection rises. Second, the relationship between the force of spillover and average age at infection is weakened by waning immunity ($\omega$). We will see in the next sections that these simple results—when combined with an increased likelihood of progressing to clinical disease with advancing age at infection—help define conditions where reducing the force of spillover can increase the overall public health burden of zoonotic disease.

## Age-specific disease severity can undermine public health benefits of spillover reduction programs

For many infectious diseases, clinical severity increases with age at infection. This has been demonstrated for Lassa virus [35, 37], SARS-CoV-2 [38], and a wide range of other bacterial and viral pathogens [18]. Although the specific functional form of this increase varies with disease, we can model the basic phenomenon by assuming that the rate at which infected individuals become clinically diseased, $\mu$, increases linearly with advancing age such that:

$$\mu = \mu_0 + \alpha a \qquad (8)$$

where $\alpha$ quantifies how rapidly the rate of transition to clinical disease increases with advancing age.

Eq (8) can be combined with steady-state solutions for the age distribution of infected individuals to approximate the burden of infectious disease, $\mathcal{B}$, as long as transitions to clinical

disease are rare (S1 Text). Analyzing the relationship between burden and the force of spillover reveals that, in some cases, burden may peak for intermediate values of spillover, creating the potential for spillover reduction to actually increase the health burden of zoonotic disease (Fig 2). If immunity is lifelong ($\omega = 0$), this possibility arises anytime:

$$\alpha^* > \frac{\mu_0 \delta (\gamma + \delta)}{\gamma} \tag{9}$$

Specifically, if condition (9) holds, the health burden of spillover is maximized at intermediate values of the force of spillover (Fig 2, solid lines). This condition is most likely to be satisfied when the rate of transition to clinical disease increases rapidly with age. Numerical solutions of the exact model verify the accuracy of the analytical approximation (Fig 2, dots). Results derived in the S1 Text show that this condition also holds if expected human lifespan is used as the relevant public health metric. Specifically, if condition (9) holds, expected human lifespan is minimized for intermediate values of the force of spillover.

If immunity wanes over time, spillover reduction is much less likely to negatively impact human health (Fig 3). Specifically, if immunity wanes over time ($\omega > 0$), the health burden of zoonotic spillover is maximized for intermediate values of the force of spillover only when a

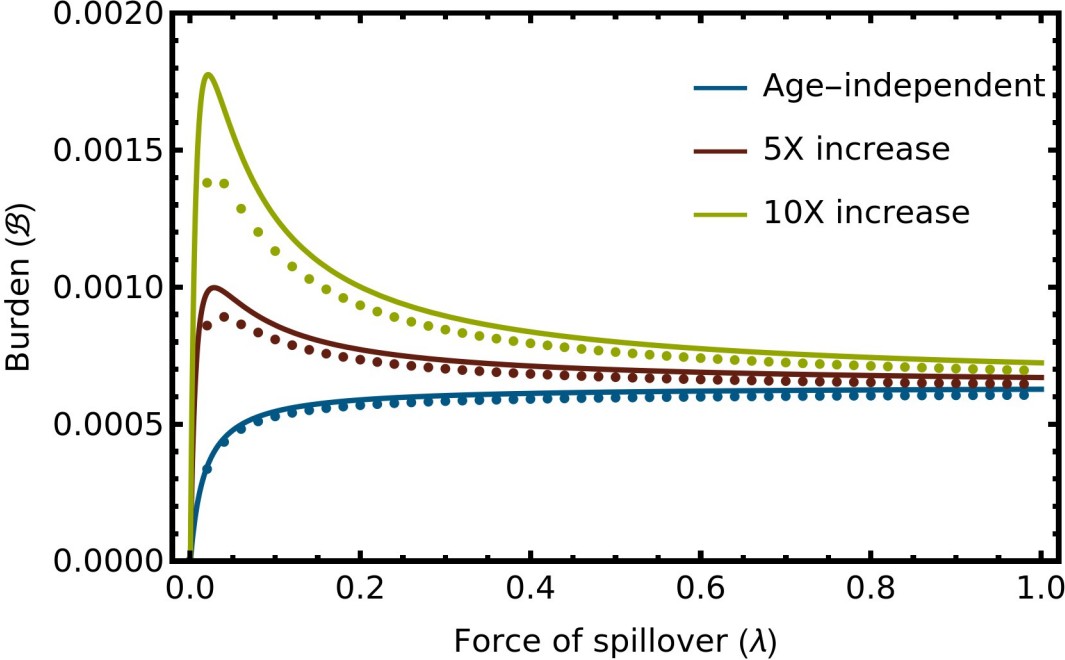

**Fig 2. The health burden of zoonotic disease, $\mathcal{B}$, as a function of the force of spillover ($\lambda$) for disease severity that increases at different rates with advancing age.** As the severity of disease increases more rapidly with advancing age, the scope for spillover reduction to negatively impact human health grows. In this example, the rate at which infected individuals become diseased increases following Eq (8) with the intercept set to $\mu_0 = 1.0$ and the slope set such that the rate of transition to disease is independent of age (blue line; $\alpha = 0$), increases 5-fold (red line; $\alpha = 1/15$), or increases 10-fold (yellow line; $\alpha = 3/20$) from the time of birth to age at which an individual reaches their expected natural lifespan ($1/\delta$). The dotted lines are numerical solutions to the exact model that do not assume the rate of transition to clinical disease is rare. The analytical approximations slightly overestimate the clinical burden because individuals leaving the $I$ class as they become diseased are ignored. This gap between approximation and exact solution grows as the rate of progression to clinical disease ($\mu$) increases, although the general shape of the curves remains consistent. The remaining parameters were: $b = 100$, $\delta = 1/60$, $\nu = 1$, $\gamma = 365/14$.

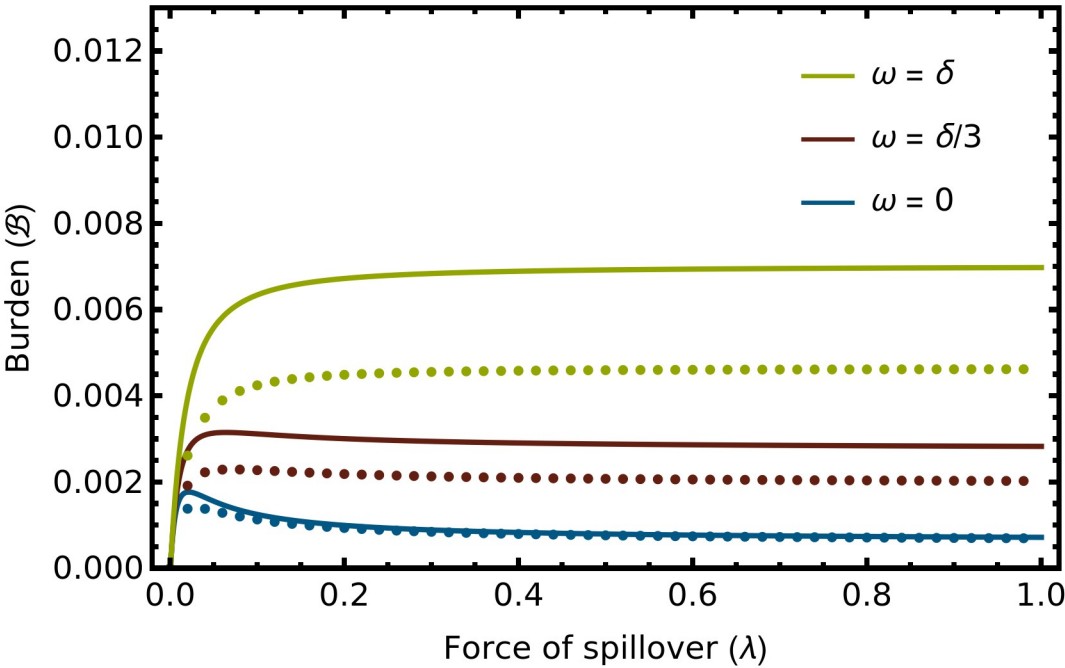

**Fig 3. The health burden of zoonotic disease, $\mathcal{B}$, as a function of the force of spillover ($\lambda$) for immunity that wanes at three different rates.** As immunity becomes more transient, the scope for spillover reduction to negatively impact human health shrinks. The blue line is a point of reference and shows the case where immunity is lifelong and spillover reduction can negatively impact human health. The red line shows a case where immunity wanes extremely slowly, lasting on average, three times the expected human lifespan. In this case, negative impacts can still occur, but they are extremely weak. Finally, the green line shows a case where immunity wanes sufficiently rapidly for negative impacts on human health to no longer be possible. Remarkably, this occurs even though this green line illustrates a scenario where immunity still lasts, on average, for the average lifespan of the human population. The dotted lines are numerical solutions to the exact model that do not assume the rate of transition to clinical disease is rare, but instead explicitly track the movement of individuals from the $I$ class into the $M$ class. The analytical approximation overestimates the clinical burden as in Fig 2, but here the discrepancy between the analytical prediction and the numerical solution to the exact model becomes more appreciable as immunity wanes more rapidly. This occurs because waning immunity increases the proportion of the population in the diseased state, and this state is ignored by our analytical approximation. In this example, the rate at which infected individuals become diseased increases following Eq (8) with the intercept set to $\mu_0 = 1.0$ and the slope set to $\alpha = 3/20$ such that the rate of transition to disease increases 10-fold from the time of birth to the age at which an individual reaches their expected natural lifespan ($1/\delta$). The remaining parameters were: $b = 100$, $\delta = 1/60$, $\nu = 1$, $\gamma = 365/14$.

set of restrictive conditions hold:

$$\alpha > \frac{\delta\mu_0(\gamma+\delta)(\delta+\omega)(\gamma+\delta+\omega)}{\gamma(-\omega(\gamma+2\delta)+\delta(\gamma+\delta)-\omega^2)} \tag{10}$$

and either inequality (11) or inequalities (12) are satisfied:

$$\delta \geq \omega(1+\sqrt{2}) \tag{11}$$

$$\omega < \delta < \omega(1+\sqrt{2}) \tag{12a}$$

$$\delta > \omega + \frac{\sqrt{\gamma^2+8\omega^2}-\gamma}{2} \tag{12b}$$

The first condition (10) requires that the rate of transition to clinical disease increases rapidly with age and is generally more restrictive than the parallel condition with lifelong immunity

(9). The additional conditions (11 and 12) require that immunity wanes slowly relative to lifespan. The reason waning immunity has such a significant impact, of course, is that it prevents immunity from becoming permanently established at a young age when the health consequences of infection are mild. Thus, even individuals infected early in life can be infected again later in life when the likelihood of clinical disease is increased. As a consequence, reducing the force of spillover generally reduces the public health burden of zoonotic disease in an intuitive way when immunity wanes over time.

## The ultimate consequences of spillover reduction take years to materialize

Up until this point, our results have focused on steady-state solutions and ignored the temporal dynamics of a transition from a state where the force of spillover is high to one where the force of spillover has been reduced, but not entirely eliminated. To better understand the consequences of partial spillover reduction for human health during the period of transition, we used stochastic simulations. Simulations were established with a burn-in period of 200 years during which the population was exposed to a relatively high force of spillover. After this initial burn-in period, the population was tracked for an additional 200 years during which the force of spillover remained high before reducing the force of spillover in year 200 and then tracking the population for another two hundred years under the reduced force of spillover regime. Simulations were performed over a broad range of parameter combinations and the resulting temporal patterns of infection and health burden were observed.

Stochastic simulations supported our steady-state solutions and analytical approximations, but also identified important transient features. For instance, Fig 4 shows a case where the analytical condition (9) is satisfied. As expected, the health burden of spillover ultimately increases in response to the reduction in the force of spillover, and the average lifespan decreases. In contrast, Fig 5 shows a case where the analytical condition (9) is not satisfied and the health burden of spillover fails to increase in response to the reduction in the force of spillover as expected from our analytical prediction. These figures also demonstrate two additional results that proved to be quite general in our simulation analyses. First, when negative health impacts do occur, they take years to materialize, generally on the order of the average human lifespan. In the years following an abrupt reduction in the force of spillover, the population may enjoy a deceptive reprieve before the health burden of spillover increases to levels exceeding historical precedent. Second, even when negative health impacts do not occur (e.g., Fig 5), the benefits of spillover reduction may be transient. For instance, even though spillover reduction has positive health impacts in Fig 5, as predicted, the public health benefits enjoyed by the population immediately following spillover reduction erode significantly over time. This erosion occurs through the same basic mechanism that enables negative impacts: as the force of spillover falls, the age at infection rises, and more individuals are infected later in life when severe disease is more likely to occur.

To evaluate the generality of our results, we explored additional simulations that relaxed the assumption of lifelong immunity. These simulations support our analytical predictions that negative consequences of spillover reduction become less likely if immunity wanes (Fig 6). In addition, these simulations demonstrate that waning immunity stabilizes the health benefits gained through partial spillover reduction (Fig 6). For instance, when immunity lasts, on average, for only 1/4 the expected human lifespan, our simulations show that the initial public health benefits of partially reducing spillover are more stable and only mildly eroded over time (Fig 6). Here too, the importance of waning immunity stems from its role in decoupling the force of spillover from the average age of infection. This occurs because waning immunity allows individuals that were previously infected earlier in life to be reinfected later in life,

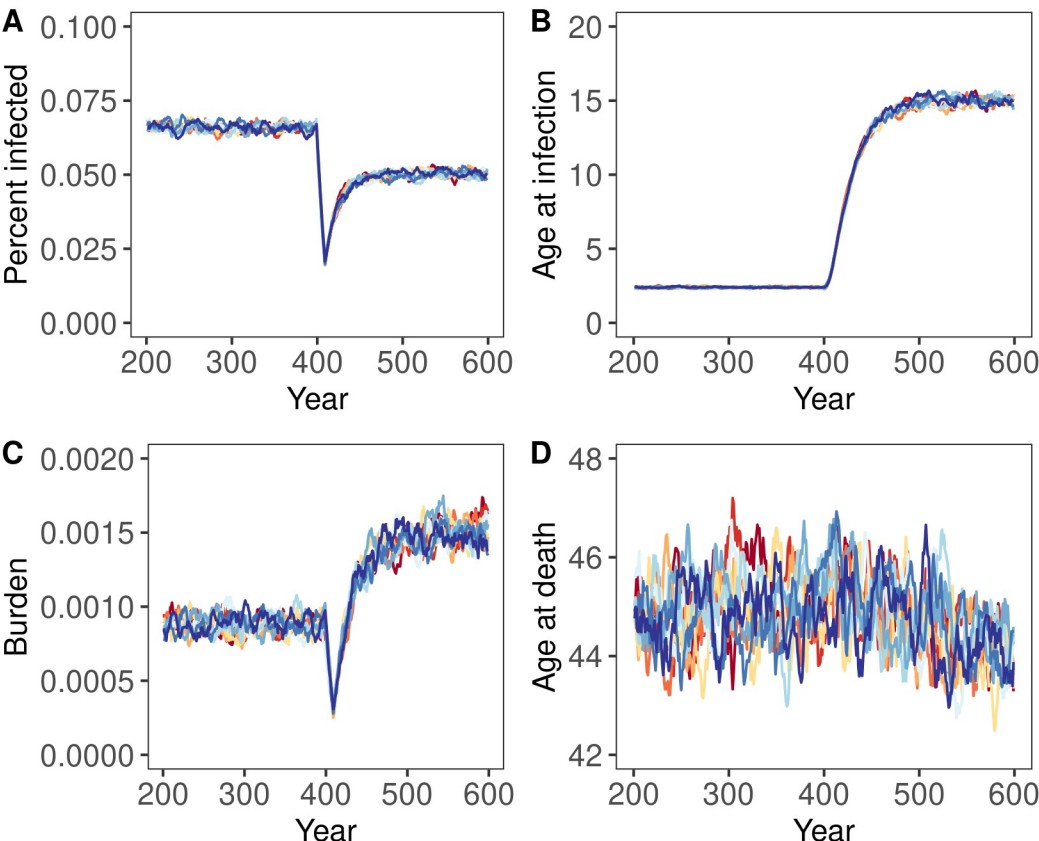

**Fig 4. The temporal dynamics of infection before (years 200–400) and after (years 401–600) an 87.5% drop in the force of spillover for a case with lifelong immunity and where negative impacts of spillover reduction are predicted to occur.** Each panel shows the output from ten replicate simulation runs, with each line showing a ten year rolling average of the value for each individual simulation run. Panel A shows the percentage of the human population with active infection, Panel B depicts the average age at which individuals are infected, Panel C reports the burden of disease, $\mathcal{B}$, and Panel D shows the average human lifespan. Parameters were: $\alpha = 0.15$, $\mu_0 = 1.0$, $b = 500$, $\delta = 1/60$, $\nu = 365/7$, $\gamma = 365/14$.

weakening the relationship between the force of spillover and average age of infection. Although waning immunity makes negative consequences of spillover reduction less likely, the burden of spillover is actually greater with waning immunity due to the additional disease burden imposed by reinfection.

## Application: Lassa virus in West Africa

Using parameter values estimated for Lassa virus allows us to evaluate if spillover reduction could potentially cause a counterintuitive and harmful increase in the health burden of this zoonotic disease. Specifically, if we assume that immunity is lifelong, the critical condition (9) evaluated for parameter values estimated for Lassa virus demonstrates that spillover reduction could conceivably increase the health burden of this disease. In contrast, if we assume immunity wanes at the rate estimated by [34], the critical conditions (10, 11, and 12) predict only positive health impacts of spillover reduction.

To better understand the likely consequences of spillover reduction at the more granular scale of individual populations, we calculated the force of spillover for 94 different human populations for which systematic serosurveys were conducted and compiled by [39]. Specifically,

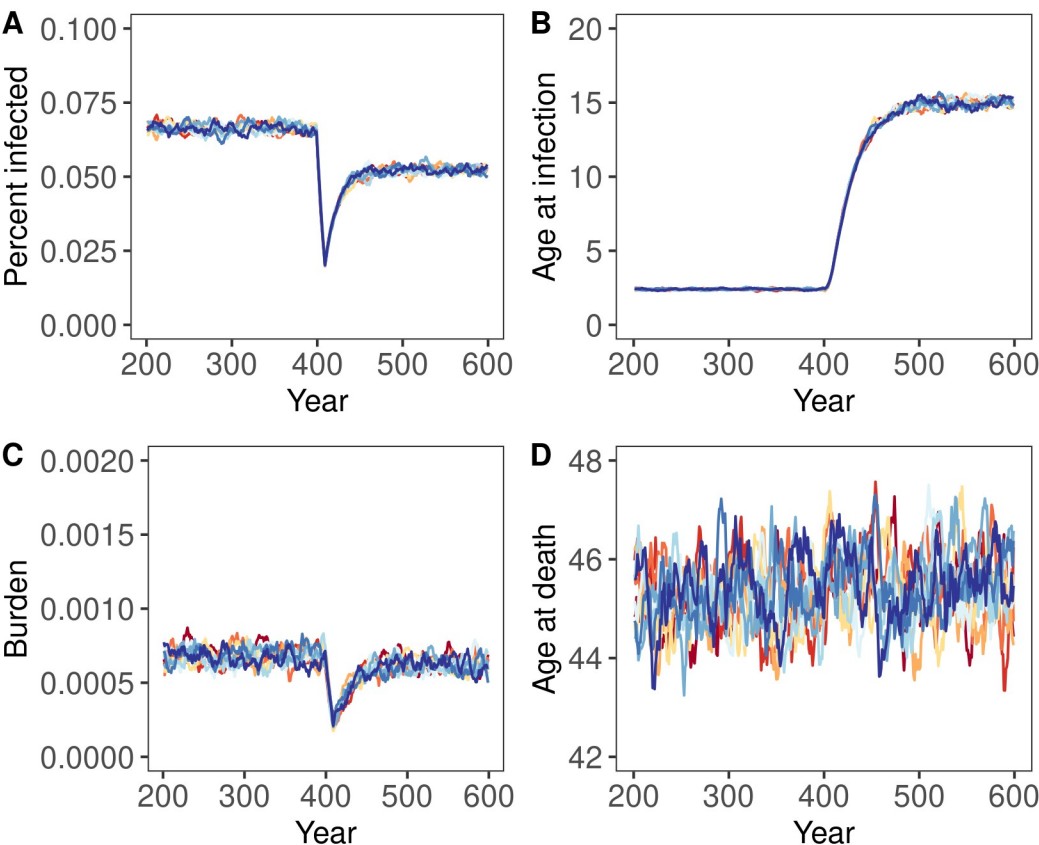

**Fig 5. The temporal dynamics of infection before (years 200–400) and after (years 401–600) an 87.5% drop in the force of spillover for a case with lifelong immunity and where negative impacts of spillover reduction are not predicted to occur.** Each panel shows the output from ten replicate simulation runs, with each line showing a ten year rolling average of the value for each individual simulation run. Panel A shows the percentage of the human population with active infection, Panel B depicts the average age at which individuals are infected, Panel C reports the burden of disease, $\mathcal{B}$, and Panel D shows the average human lifespan. Parameters were: $\alpha = 0.0167$, $\mu_0 = 1.0$, $b = 500$, $\delta = 1/60$, $\nu = 365/7$, $\gamma = 365/14$.

we used Eq (5) to predict the force of spillover for each site where seroprevalence of Lassa virus antibodies had been systematically evaluated (S1 Text). We also predicted the force of spillover for three additional hypothetical sites with levels of Lassa virus seroprevalence greater than any populations in our data set and equal to 0.65, 0.75, and 0.85. Next, we predicted the equilibrium health burden as a function of the force of spillover using numerical solutions of the ordinary differential equations (S1 Text equations S1) and numerical integration of Eq (3). Finally, we placed each site on this curve at the position corresponding to its estimated force of spillover. This analysis demonstrates that even when immunity is lifelong, the force of spillover estimated for the sites in our study places them to the left of the peak health burden (Fig 7; red dots). As a consequence, reducing the force of spillover within these sites would have positive impacts on human health. Only the hypothetical site with Lassa virus seroprevalence equal to 0.85 could experience a negative health impact of spillover reduction (Fig 7; blue dots). We evaluated the sensitivity of these results by studying additional scenarios where the rate of transition to severe disease increased more rapidly than we estimate. Specifically, we studied cases where the slope of the relationship was 1.5 and 2.0 times the value we estimated using the data from [35]. The results of these additional analyses demonstrate that our general conclusion remains robust—even when the rate of transition to clinical disease is twice as sensitive to

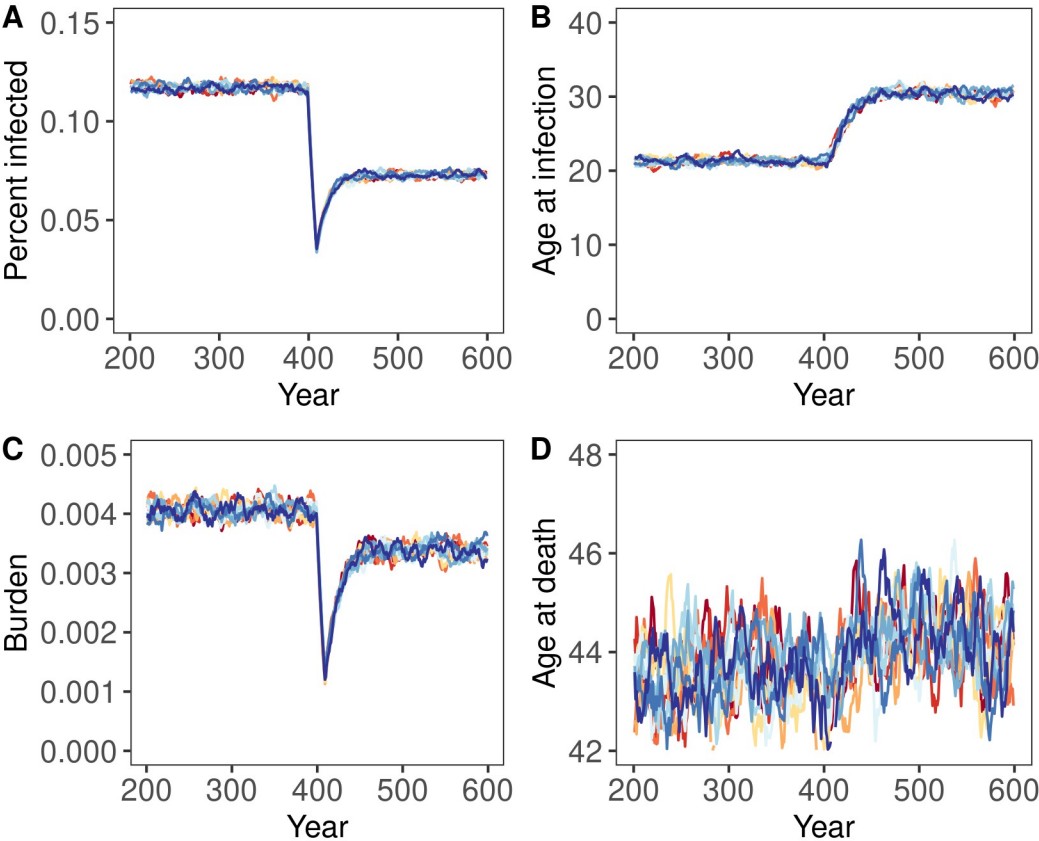

**Fig 6. The temporal dynamics of infection before (years 200–400) and after (years 401–600) an 87.5% drop in the force of spillover for a case with slowly waning immunity.** Parameters were identical to those used in Fig 4 except that immunity lasts, on average, only as long as the expected human lifespan ($\omega = 1/60$). As in Fig 4, each panel shows the output from ten replicate simulation runs, with each line showing a ten year rolling average of the value for each individual simulation run. Panel A shows the percentage of the human population with active infection, Panel B depicts the average age at which individuals are infected, Panel C reports the burden of disease, $\mathcal{B}$, and Panel D shows the average human lifespan. Parameters were: $\alpha = 0.0167$, $\mu_0 = 1.0$, $b = 500$, $\delta = 1/60$, $\nu = 365/7$, $\gamma = 365/14$.

advancing age as our estimate, no populations in our dataset are predicted to experience negative consequences of spillover reduction (S1 Text).

## Discussion

We have developed and analyzed mathematical and computational models to identify conditions where reducing the spillover of zoonotic pathogens can have unanticipated, negative impacts on human health. Our models focus on zoonotic pathogens that infect humans primarily through direct spillover with little or no subsequent human-to-human transmission. We use these models to clarify when increases in the average age at infection caused by a reduction in the force of spillover can increase overall levels of disease within the human population. Model analyses demonstrate that negative public health impacts of spillover reduction are generally unlikely and that two restrictive conditions must be met for negative consequences to arise. First, disease severity must increase sufficiently rapidly with advancing age. Second, pathogen immunity must be durable and wane only very slowly over time. Although our results suggest that negative impacts of spillover reduction are likely to be rare, they also demonstrate that the benefits of spillover reduction may dissipate over time as the system

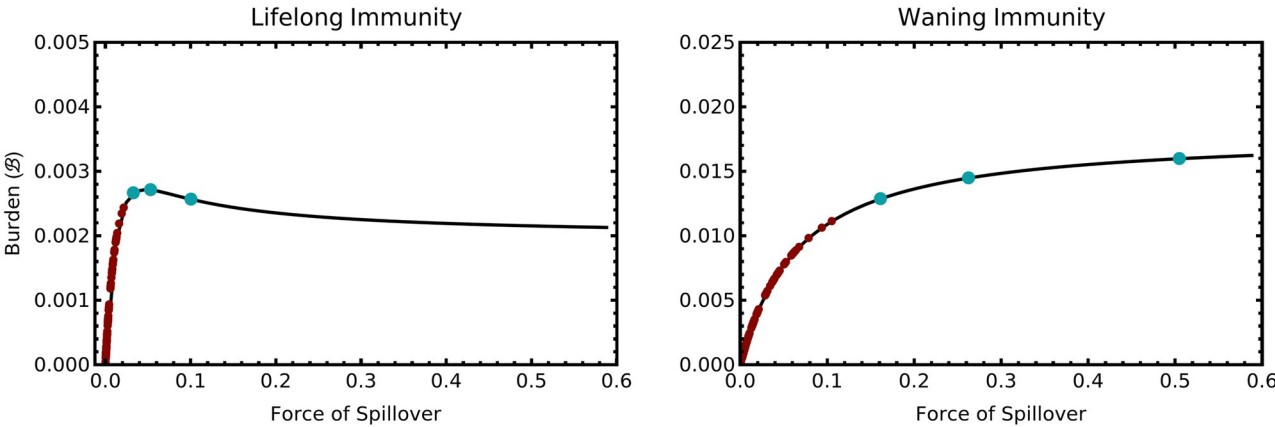

**Fig 7. The predicted disease burden of Lassa virus infection for the case of lifelong immunity (left panel) and waning immunity (right panel).** The black line is the theoretical prediction for each case as a function of the force of spillover. The red dots show the force of spillover estimated for actual sites in West Africa where systematic serosurveys have been conducted. The blue dots show hypothetical populations with a force of spillover estimated for seroprevalences of $\mathcal{R} = 0.65$, $\mathcal{R} = 0.75$, and $\mathcal{R} = 0.85$. For the case of waning immunity, the expected duration of immunity is set to 15.63 years ($\omega$ = 0.064) as estimated by [34]. Note that the burden of zoonotic disease is significantly greater with waning immunity because reinfection is possible. Parameter values were as described in the S1 Text. Birth rate of the human population was set to $b$ = 24.75 which yields a local population size of 1500.

settles into a new equilibrium state. These results suggest that the beneficial public health effects of spillover reduction will be greatest and last the longest when a near-complete elimination of spillover risk can be achieved or when immunity to a pathogen is transient and reinfection common.

While spillover reduction is unlikely to negatively impact human health, negative impacts are possible if disease severity increases sufficiently rapidly with age. Accumulating studies of the relationship between these quantities suggest that significant increases in disease severity with advancing age are common [18], but how often disease severity increases sufficiently rapidly for negative consequences to be widespread remains unclear. Additional uncertainty arises because of sensitivity to the dynamics of waning immunity and reinfection. Specifically, in our model, immunity to the pathogen must be very durable and reinfection unlikely for negative consequences to be possible. The importance of durable immunity in our model stems from the role reinfection plays in breaking associations between spillover pressure and the average age at infection. Specifically, because waning immunity returns previously infected individuals to the susceptible class, it weakens the association between infection and age. Thus, in cases where immunity wanes rapidly, reducing spillover decreases the force of infection but has little impact on the average age of infection, undermining the mechanism through which counterintuitive increases in disease burden arise. An important caveat to this result is that we assume secondary infections (i.e., those acquired by previously infected individuals who have recovered and subsequently lost immunity) are associated with the same risk of developing severe disease as are primary infections. If, instead, secondary infections are less likely to develop severe disease (e.g., because of a long-lasting T-cell response), waning immunity will do less to guard against the possibility that spillover reduction increases disease burden than our results suggest. Because the relationship between reinfection and disease severity is poorly understood [40], a conservative approach to assessing risk is to assume lifelong immunity.

When our model is parameterized for Lassa virus, our results suggest negative consequences of spillover reduction efforts should be possible only in populations where the current force of spillover is extremely high. Specifically, none of the populations for which we could find seroprevalence estimates are predicted to experience a force of spillover sufficient for

spillover reductions to lead to negative health consequences (Fig 7). This result holds even if the rate at which disease severity increases with advancing age is twice the value we estimate using data from [35]. Although these results are encouraging, populations may well exist where the contemporary force of Lassa virus spillover exceeds the threshold value required for negative consequences. This leads to the disconcerting possibility that the populations most likely to be targeted for spillover reduction efforts because of high seroprevalence and infection rates may also be those most likely to experience negative impacts of partial spillover reduction. An additional complication in the Lassa virus system is that the force of spillover may be decreasing naturally as an invasion by the black rat, *Rattus rattus*, displaces the primary rodent reservoir of Lassa virus, *Mastomys natalensis* [28–31]. Our results show that this reduction in spillover could, in principle, change the dynamic of human infection from one defined by frequent mild childhood infections to one defined by infrequent but severe infections in older individuals.

Although our model is quite general, it makes a range of assumptions that could influence the likelihood that spillover reduction causes negative public health impacts. For instance, our results are predicated upon an absence of human-to-human infection and thus are directly applicable to only a subset of zoonotic pathogens. It seems likely, however, that our results would continue to apply in cases where human-to-human transmission following initial spillover is weak and not self-sustaining ($R_0 < 1$). In such cases, human-to-human transmission would simply result in modest increases in the force of infection which should have little impact on our overall results. Another important assumption of our model is that the force of spillover is equal across human age classes. This assumption was made for simplicity and is likely to be violated when exposure to wild animals carrying zoonotic pathogens varies significantly with age. For instance, recent work suggests that exposure risk to Lassa virus may be concentrated in young children as a consequence of hunting wild rodents [41]. Because age-specific patterns of exposure are likely to be idiosyncratic and system specific, however, it is not clear what consequence they will have for the outcome of spillover reduction programs. Finally, our applied results on Lassa virus assume human populations have reached a steady-state where the force of spillover has been constant for long enough that the age structure of immunity has stabilized. Although this assumption is convenient and difficult to relax given the limited data at our disposal, it is unlikely to hold because of extensive recent human migration within West Africa [42] and because of changes in the distribution and abundance of the rodent reservoir caused by an ongoing biological invasion [31].

Reducing the spillover of zoonotic pathogens holds incredible promise for improving human health [14–17]. Our results demonstrate, however, that this approach is not entirely without risk in systems where the severity of infection increases with age. Developing a better understanding of the relationship between disease severity and age at infection and the extent to which this relationship holds for reinfection, will help clarify if the risks are minimal, as our results suggest for Lassa virus, or can become appreciable in some systems. From a practical perspective, our results suggest two guidelines for disease mitigation that minimize the risk of adverse consequences and provide projections of public health gains. First, before implementing interventions that reduce spillover, system-specific models parameterized with the best available data should be used to quantify benefits and risks over various time horizons and implementation strategies. These models can be used to set realistic expectations for possible outcomes and allow affected communities to make informed decisions about the merits of proposed spillover control programs. Second, interventions should be selected that have a high probability of eliminating most spillover risk. Interventions that partially reduce spillover but allow it to remain appreciable create the greatest scope for adverse public health outcomes and increase the likelihood that public health gains will be short-lived. Pursuing well-planned

spillover reduction programs that follow these guidelines remains one of the most promising methods available for dealing with the long-standing challenge of managing zoonotic disease.

## Disclaimer

The content is solely the responsibility of the authors and does not necessarily represent the official views of the National Institutes of Health or the National Science Foundation.

## Supporting information

**S1 Text. Supporting derivations and data.** This file includes complete derivations of analytical results, descriptions of supporting numerical analyses, descriptions of supporting data, and supporting figures.
(PDF)

## Author Contributions

**Conceptualization:** Scott L. Nuismer, Andrew J. Basinski, Elisabeth Fichet-Calvet, Christopher H. Remien.

**Formal analysis:** Scott L. Nuismer, Andrew J. Basinski, Christopher H. Remien.

**Funding acquisition:** Scott L. Nuismer.

**Writing – original draft:** Scott L. Nuismer, Evan A. Eskew, Christopher H. Remien.

**Writing – review & editing:** Scott L. Nuismer, Andrew J. Basinski, Courtney L. Schreiner, Evan A. Eskew, Elisabeth Fichet-Calvet, Christopher H. Remien.

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
