## [Decision Letter · Decision Letter 0]

25 Apr 2024

Dear Dr. Nuismer,

Thank you very much for submitting your manuscript "Quantifying the risk of spillover reduction programs for human health" for consideration at PLOS Computational Biology.

As with all papers reviewed by the journal, your manuscript was reviewed by members of the editorial board and by several independent reviewers. In light of the reviews (below this email), we would like to invite the resubmission of a significantly-revised version that takes into account the reviewers' comments.

Reviewers emphasized the need for a more nuanced characterization of the conditions under which spillover translates to negative public health consequences in real-world scenarios. Both reviewers offered valuable suggestions for strengthening this aspect, proposing ideas to explore further aspects linking theoretical concepts and practical applications.

We cannot make any decision about publication until we have seen the revised manuscript and your response to the reviewers' comments. Your revised manuscript is also likely to be sent to reviewers for further evaluation.

Sincerely,

Claudio José Struchiner, M.D., Sc.D.

Academic Editor

PLOS Computational Biology

Thomas Leitner

Section Editor

PLOS Computational Biology

Reviewers emphasized the need for a more nuanced characterization of the conditions under which spillover translates to negative public health consequences in real-world scenarios. Both reviewers offered valuable suggestions for strengthening this aspect, proposing ideas to explore further aspects linking theoretical concepts and practical applications.

Reviewer's Responses to Questions

**Comments to the Authors:**

Reviewer #1: In this manuscript, the authors test whether reduction of spillover can have unexpected negative consequences in terms of disease burden in scenarios where the risk of severe disease increases with age. The basic idea is that reducing spillover pushes the age of first infection later, and thus can increase average severity. They use mathematical models (simulations and numerical solutions for a PDE model) and find that when immunity does not wane, age related increases in disease severity can lead to increased disease burden as spillover is reduced. The authors illustrate the parameter space in which this outcome arises, and then parameterizes their model for Lassa virus. For Lassa, such effects are not seen at observed levels of spillover, but would be seen if spillover rates were higher. This idea was interesting, well presented and easy to understand, but I was left wondering how often (if ever) decreasing spillover would lead to increased disease burden in the real world. I am therefore left with the impression that these patterns are fascinating in theory; but their practical relevance is unclear (and overemphasized in the discussion). To this point, we express the following concerns:

One of my primary concerns is that the manuscript currently leaves the reader with the message that spillover reduction is something that should be approached with caution, but the results don't seem to align with this perspective. It is not clear how often reduction of spillover will actually result in negative public health consequences in practice, despite being presented as a significant risk. Perhaps the authors could point to examples of real systems where these negative consequences may be likely, or alternatively ease up on the message that reducing spillover can be bad -- I would probably suggest the latter given that the authors have already shown that even in the worst cases, reducing spillover leads to short term benefits (akin to vaccine honeymoon periods).

Furthermore, observing negative public health consequences could depend on the metric used to measure public health burden. The authors currently use the number of severe disease cases. While this is reasonable, other metrics seem more relevant to me. For example, does reducing spillover also reduce life expectancy or increase DALYs (disability adjusted life years)? As we learned from COVID, the health of elderly individuals is not always valued in the same way as the health of children/workers.

Minor Comments:

It was shown that disease burden could be maximized for intermediate spillover only if the relationship between age and severity was sufficiently strong. Since this suggests that burden would be driven by infections of older individuals in the population, it might be useful to show the distribution of ages at infection rather than just the average age of infection in Figs 4 and 5. This would probably become even more important if risk increased non-linearly with age, as it likely does, which could be worth discussing.

Reducing spillover seemingly reduces overall infection prevalence (percent infected, as shown in figures 4, 5) regardless of the impact on the metric of disease burden. The model does not currently incorporate any human-to-human transmission. Perhaps the authors could speculate on how limited human transmission (i.e., R_0<1) might impact the change in public health burden as spillover is reduced, so that this theory could be applied more generally.

The ecological driving mechanisms of spillover can result in inhomogeneous exposure over time and between different age groups, and this could have significant implications on the metric for disease burden. Namely, if exposure risk and disease severity are inversely related (e.g. children more likely to come in contact with wildlife), reducing spillover would be unlikely to increase burden since high-risk individuals are unlikely to be exposed regardless. It would be useful to at least consider how age-dependencies in exposure risk and disease severity might interact within this framework.

Is the maximal burden for intermediate levels of spillover only possible with linear relationships, or would this trend hold for any function that is sufficiently quickly increasing. Could other shapes possibly yield a similar trend (i.e., high risk in very young and very old individuals)?

The parameter choice for Figure 3 seemed strange: Why were these particular values used when they hardly show increased burden for intermediate spillover?

In Figures 4 & 5, perhaps consider adding mean lines from several simulations instead of just points from a single replicate. This could help demonstrate consistency between simulation replicates. It does not seem relevant in this particular context, but this does not necessarily show if some simulations reach different equilibria under the same parameter set.

I was curious about the observation that basically any waning removed the observed pattern. Is this because of the assumption that waning is exponential. Would simulations with discrete waning periods or multiple recovered classes (yielding gamma distributed wait times) “recover” this pattern so to speak?

Typo in Figure 2 key: “Age-independent”

I really don't like the term "exact numerical solution" in the Figure 2 legend. I suppose there are some cases where a numerical solution can be exact, but is that the case here? Perhaps “numerical solution to the exact model” would be more accurate.

Reviewer #2: Referee report on PCOMPBIOL-D-24-00181

Quantifying the risk of spillover reduction programs for human health

SUMMARY

The paper uses simulations of a PDE model to explore the implications of the following theoretical mechanism. Assuming plausible demographics, interventions designed to decrease the force of zoonotic spillover would increase the average age of infection in the population. As a result, programs that reduce spillover can theoretically increase disease burden. This theoretical possibility requires that disease severity sufficiently increases with age, and that reinfection are sufficiently unlikely. Thus, primary prevention measures targeting spillover reduction have theoretically unambiguous beneficial effects only when they can eliminate the spillover risk or in cases where loss of immunity and reinfection are likely.

As an application, the model is parameterized to the case of the Lassa virus in West Africa. In that case, the theoretical argument about the negative consequences of spillover reduction efforts would be relevant only for populations experiencing an extremely high current spillover force. As a result, none of the actual populations for which the authors have sero-prevalence estimates satisfy this condition.

The paper is tightly-focused and well-executed. I only have some minor comments below.

MINOR COMMENTS

(1) Lines 57-59: “Our models apply to those zoonotic diseases caused primarily by repeated, direct spillover from an animal reservoir rather than an initial spillover followed by sustained human-to-human transmission.”

It would be interesting if the authors briefly discussed in their conclusion section the assumption of a sub-critical pathogen throughout the paper. It does not seem that the main theoretical argument depends on this assumption and the super-critical case could be an interesting topic for a sequel paper.

(2) Lines 134-138 (and appendix): “Assuming the population is at a steady state…”

This assumption is used in the estimation, as well as in different parts of the paper. Does this assumption hold in the data? It should not be hard to check whether the population under study experiences population growth and/or whether the demographic characteristics are changing over the sample period.

(3) Line 146: “we calculate the average age at infection for our model”

Related to the above equation, does the theoretical average infection age match the infection pattern observed in the data? It might be useful to comment on this since, as shown later in paper, the steady state assumption is not entirely innocuous.

**Have the authors made all data and (if applicable) computational code underlying the findings in their manuscript fully available?**

Reviewer #1: Yes

Reviewer #2: None

PLOS authors have the option to publish the peer review history of their article (what does this mean?). If published, this will include your full peer review and any attached files.

Reviewer #1: No

Reviewer #2: No
---

## [Decision Letter · Decision Letter 1]

22 Jul 2024

Dear Dr. Nuismer,

We are pleased to inform you that your manuscript 'Quantifying the risk of spillover reduction programs for human health' has been provisionally accepted for publication in PLOS Computational Biology.

Best regards,

Claudio José Struchiner, M.D., Sc.D.

Academic Editor

PLOS Computational Biology

Thomas Leitner

Section Editor

PLOS Computational Biology

Reviewer's Responses to Questions

**Comments to the Authors:**

Reviewer #1: The authors have done a great job addressing all of my concerns.

Reviewer #2: I have now read the revised version of the paper and find it ready for publication. The authors did a good job in addressing my earlier comments, or in convincingly arguing why some of them could not be readily addressed within their current framework.

**Have the authors made all data and (if applicable) computational code underlying the findings in their manuscript fully available?**

Reviewer #1: None

Reviewer #2: Yes

PLOS authors have the option to publish the peer review history of their article (what does this mean?). If published, this will include your full peer review and any attached files.

Reviewer #1: No

Reviewer #2: No

---

## [Editor Report · Acceptance letter]

9 Aug 2024

PCOMPBIOL-D-24-00181R1 

Quantifying the risk of spillover reduction programs for human health

Dear Dr Nuismer,

I am pleased to inform you that your manuscript has been formally accepted for publication in PLOS Computational Biology. Your manuscript is now with our production department and you will be notified of the publication date in due course.

With kind regards,

Anita Estes
